# Distal Symmetric and Cardiovascular Autonomic Neuropathies in Brazilian Individuals with Type 2 Diabetes Followed in a Primary Health Care Unit: A Cross-Sectional Study

**DOI:** 10.3390/ijerph17093232

**Published:** 2020-05-06

**Authors:** Mozania Reis de Matos, Daniele Pereira Santos-Bezerra, Cristiane das Graças Dias Cavalcante, Jacira Xavier de Carvalho, Juliana Leite, Jose Antonio Januario Neves, Sharon Nina Admoni, Marisa Passarelli, Maria Candida Parisi, Maria Lucia Correa-Giannella

**Affiliations:** 1Programa de Pos-Graduaçao em Medicina, Universidade Nove de Julho (UNINOVE), Rua Vergueiro 235, 2° subsolo, Pos-graduação, Sao Paulo 01504-001, Brazil; moszania@gmail.com (M.R.d.M.); julianaleitet@gmail.com (J.L.); joseajneves@gmail.com (J.A.J.N.); m.passarelli@uni9.pro.br (M.P.); 2Unidade Basica de Saude Dra. Ilza Weltman Hutzler. Rua Coronel Walfrido de Carvalho, Sao Paulo 02472-180, Brazil; crisdiaslucas@gmail.com (C.d.G.D.C.); xavierjacira@gmail.com (J.X.d.C.); 3Laboratorio de Carboidratos e Radioimunoensaio (LIM-18), Hospital das Clinicas HCFMUSP, Faculdade de Medicina da Universidade de Sao Paulo. Avenida Dr. Arnaldo, 455, Sala 3321, Sao Paulo 01246-903, Brazil; dps.daniele@usp.br (D.P.S.-B.); sharonadmoni@gmail.com (S.N.A.); 4Laboratorio de Lipides (LIM-10), Hospital das Clinicas HCFMUSP, Faculdade de Medicina da Universidade de Sao Paulo, Avenida Dr. Arnaldo, 455, Sala 3305, Sao Paulo 01246-903, Brazil; 5Divisao de Endocrinologia, Departamento de Clinica Medica, Faculdade de Medicina da Universidade Estadual de Campinas (UNICAMP), Rua Tessalia Vieira de Camargo, 126 – Cidade Universitaria, Campinas 13083-887, Brazil; candidap@unicamp.br

**Keywords:** diabetic neuropathy, distal symmetric polyneuropathy, primary care, cardiovascular autonomic neuropathy

## Abstract

The paucity of epidemiological data regarding diabetes complications in Brazil motivated us to evaluate the prevalence rates of distal symmetric polyneuropathy (DSP) and of cardiovascular autonomic neuropathy (CAN) in individuals with type 2 diabetes (T2D) followed in a primary care unit. A total of 551 individuals (59.3% women, 65 years old; diabetes duration of 10 years; HbA1c of 7.2%, medians) were included in this cross-sectional study. DSP was diagnosed by sum of the Neuropathy Symptoms Score (NSS) and Modified Neuropathy Disability Score (NDS) and by the Semmes–Weinstein monofilament. CAN was diagnosed by cardiovascular autonomic reflex tests combined with spectral analysis of heart rate variability. The prevalence rates of DSP were 6.3% and 14.3%, as evaluated by the sum of NSS and NDS and by the Semmes–Weinstein monofilament, respectively. Those with DSP diagnosed by monofilament presented longer diabetes duration, worse glycemic control and a higher stature. The prevalence rates of incipient and definitive CAN were 12.5% and 10%, respectively. Individuals with definitive CAN presented a higher frequency of hypercholesterolemia and of arterial hypertension. The higher prevalence rate of DSP with the use of the monofilament suggests that it may be a more appropriate tool to diagnose DSP in the primary care setting in Brazil.

## 1. Introduction

Diabetic neuropathy is defined as “the presence of symptoms and/or signs of peripheral nerve dysfunction in people with diabetes mellitus after exclusion of other causes” [1,2]. Although diabetic neuropathy is considered the most common microvascular complication, there is difficulty in estimating its real prevalence. Possible causes for this difficulty are the fact that it is commonly asymptomatic, has a wide clinical spectrum, and there is a lack of homogeneity in the diagnostic criteria [1]. Distal symmetric polyneuropathy (DSP), a major cause of diabetic foot ulcers, is the most prevalent form of peripheral neuropathy, ranging from 8% to 25% in individuals with type 2 diabetes (T2D) in different studies, with a higher prevalence when asymptomatic individuals are considered (45%) [3].

Another prevalent form of diabetic neuropathy is cardiovascular autonomic neuropathy (CAN), defined as damage to the autonomic fibers responsible for heart and blood vessel innervation, which leads to dysfunction in heart rate control and in vascular dynamics [4]. The prevalence of CAN also varies according to the diagnostic criteria, and its importance lies in its association with major cardiovascular events [5] and with increased mortality [6].

There is a paucity of epidemiological data regarding diabetes complications in Brazil, especially in the primary care setting. Although the risk factors for diabetes complications are well-known, the particularities of T2D individuals followed in primary care units in Brazil have been infrequently characterized. Therefore, the objective of the present study was to evaluate the prevalence of DSP and of CAN in individuals with T2D followed in a basic health unit in the city of São Paulo, and the risk factors associated with them. Knowing the frequency of these conditions directly implicated in lower-limb amputations [3] and cardiovascular events [5], respectively, may be important for public health planning, allowing implementation of strategies to better control risks for these complications.

## 2. Methods

### 2.1. Participants and Evaluation of DSP and CAN

Among the 21,181 individuals followed in the basic health unit in the city of São Paulo, 1853 had T2D and were invited to participate in this cross-sectional study during home visits by community health agents. A total of 583 out of 1853 T2D individuals agreed to participate and 551 were included between September 2018 and February 2019; in total, 32 individuals were not included because their capillary blood glucose was >10 mmol∙L^−1^ at the day of evaluation (hyperglycemia may affect the results of the autonomic tests) (Figure 1). Thus, this study was conducted in a convenience sample, since there was no sample size calculation, and was conducted in compliance with the Declaration of Helsinki, in accordance with the institutional ethics committees (Universidade Nove de Julho, # 81249417.1.0000.5511; Secretaria Municipal da Saude de São Paulo, # 81249417.1.3001.0086). After signing informed consent, participants were assessed for demographic, clinical, and biochemical features, and for the status of DSP and of CAN, whose evaluations were performed by three trained nurses. The estimated glomerular filtration rate (eGFR) was calculated by CKD-EPI [7]. The primary care unit has no ophthalmologists in its medical staff and the screening for diabetic retinopathy is performed in secondary care units by different ophthalmologists. For that reason, data about retinopathy are not being presented. Arterial hypertension was defined as systolic/diastolic blood pressure ≥ 140/90 mmHg or use of anti-hypertensive drugs not for renal protection purposes. Hypercholesterolemia was defined as the use of statins and/or an LDL-cholesterol > 2.6 mmol∙L^−1^ (100 mg∙dL^−1^).

DSP was diagnosed by the sum of the Neuropathy Symptoms Score (NSS; participants were graded as presenting mild (3–4), moderate (5–6) or severe (7–9 points) symptoms) and Modified Neuropathy Disability Score (NDS, participants were graded as presenting mild (3–5), moderate (6–8) or severe (9–10 points) signs). The minimum criteria required for DSP diagnosis were presence of moderate symptoms (regardless of presence of signs) or presence of mild symptoms plus moderate signs [1,8]. The Semmes–Weinstein monofilament was also tested on four plantar sites in each foot (halux, 1st, 3rd and 5th metatarsus heads) and at least one insensitive site was considered as abnormal [9].

CAN was diagnosed by cardiovascular autonomic reflex tests (CARTs, also known as Ewing tests), based on the measurement of heart rate in response to deep breathing (expiration: inspiration ratio), to Valsalva maneuver, and to lying-to-standing (orthostatic test, 30:15 ratio), combined with spectral analysis of the heart rate variability (HRV) and systolic blood pressure after three minutes standing. Thus, seven variables were evaluated: (1) expiration: inspiration ratio; (2) Valsalva ratio; (3) 30:15 ratio; (4) postural change in systolic blood pressure; (5) spectral power in the low frequency band; (6) spectral power in the high-frequency band; (6) spectral power in the very low-frequency band. Diagnosis of incipient CAN was made in the presence of two altered tests, while > three altered tests were required for the diagnosis of definitive CAN [10]. Autonomic function tests were performed in the morning, in a quiet room. Participants were advised to refrain from eating, drinking coffee and smoking for at least 8 h before the tests, and to interrupt the use of caffeine and decongestants at least 8 h before the tests. Results were defined by age- and sex-based normal values for variability and cardiac reflex tests [11]. Participants receiving β-blockers were not evaluated for CAN status (*n* = 104).

### 2.2. Statistical Analyses

Results are expressed as median ± interquartile interval except where stated otherwise. Differences between groups were assessed by Pearson’s χ2-test (nominal variables) or by Wilcoxon/Mann–Whitney test (continuous variables). Prevalence odds ratios (ORs) were estimated from a multivariable logistic regression analysis. A *p* value < 0.05 was considered significant. Statistics were conducted with the JMP software (SAS Institute, Cary, NC, USA).

## 3. Results

A total of 551 individuals with T2D (59.3% women), with a median age (interquartile interval) of 65 (59–72) years old, a median diabetes duration of 10 (5–15) years and a median HbA1c of 7.2 (6.3–9.1)% were recruited. Table 1 shows the demographic, clinical and biochemical characteristics of the participants (overall and sorted according to the status of DSP as evaluated by the monofilament), as well as the frequency of comorbidities, diabetes complications and of the use of medicines. Overall, 72% of the individuals presented arterial hypertension and 72% presented hypercholesterolemia. Regarding diabetes complications, 23.5% of the individuals presented eGFR < 60 mL/min/1.73 m^2^, 14.3% presented DSP as evaluated by the Semmes–Weinstein monofilament, 12.5% presented incipient CAN and 10% presented definitive CAN. The sum of NSS and NDS diagnosed DSP in only 6.3% of the T2D individuals.

In comparison to individuals without DSP as evaluated by the monofilament, those with DSP were taller (161 vs. 165 cm, respectively; *p* = 0.008); presented longer diabetes duration (8 vs. 13 years, respectively; *p* = 0.0008); higher HbA1c values (54 vs. 65 mmol∙L^−1^, respectively; *p* = 0.02); and higher frequency of DSP evaluated by the sum of NSS and NDS (2% vs. 33%, respectively; *p* < 0.0001), of amputations (0% vs. 11.4%, respectively; *p* < 0.0001), of NPH (23% vs. 41%, respectively; *p* = 0.0006) and of regular (6% vs. 19%, respectively; *p* = 0.0001) insulins use (Table 1). 

The characteristics of the individuals according to the status of CAN are shown in Table 2. In comparison to individuals without definitive CAN, those with this complication presented higher cholesterol concentrations (5.0 vs. 5.3 mmol∙L^−1^, respectively; *p* = 0.03) and a higher frequency of hypercholesterolemia (68% vs. 84%, respectively; *p* = 0.03) and of metformin use (74% vs. 92%, respectively; *p* = 0.01). Some variables that indicate a worse metabolic profile in individuals with CAN presented borderline *p* values, such as a higher frequency of arterial hypertension, higher concentrations of triglycerides, longer diabetes duration and higher frequency of statins and fibrates use. 

In the multivariate analyses, longer diabetes duration was identified as an independent risk factor for DSP as evaluated by the monofilament (OR = 1.05; 95% confidence interval [CI] = 1.02–1.08; *p* = 0.0003) while arterial hypertension (OR = 3.25; 95% CI = 1.12–9.39; *p* = 0.016) and higher cholesterol concentrations (OR = 1.01; 95% CI = 1.0007–1.02; *p* = 0.031) were identified as independent risk factors for CAN (Table 3).

## 4. Discussion

The general scarcity of epidemiological data in Brazil motivated us to evaluate the prevalence of DSP and CAN in individuals with T2D followed in a primary care unit of the city of São Paulo. The prevalence of DSP as evaluated by the sum of NSS and NDS was low (6.3%) when compared to other Brazilian studies which used these same tools to evaluate T2D cohorts followed in tertiary centers. In the studies of Moreira et al. [12] and Santos et al. [13], 47.8% of 94 and 39.4% of 426 individuals with T2D, respectively, were diagnosed with DSP. The comparison of the characteristics of the population of the present study with the population described by Santos et al. shows that although age (65 vs. 68 years old, respectively), gender distribution (59.5% vs. 62% of women) and disease duration (7 vs. 10 years) do not appear very different, the HbA1c values (7.2% vs. 8.1%) and the proportions of individuals taking insulin (25% vs. 67.2%) indicate that, as expected, the population followed in the tertiary center presented a more severe T2D.

During the performance of the NSS, the examiners noticed that a not negligible proportion of individuals had difficulty understanding the questionnaire. This cohort is comprised of 73.5% of individuals ≥ 60 years (34% ≥ 70 years) with relatively low education levels (58.1% had received ≤ 8 years of formal education, 34% had received between 9 and 11 years and 7.9% had received ≥ 12 years of formal education). Individuals with T2D are at increased risk of mild cognitive impairment [14] and, additionally, a health literacy below adequate has been described in Brazilian individuals with T2D [15,16]. Although no cognitive function or health literacy skills assessments were performed in the present study, we hypothesized that these factors might contribute to the difficulty in comprehending the NSS.

The use of the Semmes–Weinstein monofilament showed a prevalence of DSP of 14.3%. Individuals with this microvascular complication diagnosed by this tool presented more severe diabetes (as shown by a higher frequency of insulin use) and classical risk factors for DSP, such as longer diabetes duration, worse glycemic control and a higher stature [17]. In the multivariate analysis, however, only diabetes duration was identified as an independent risk factor for this complication, which may result from the relative homogeneity of the population regarding other variables. The higher frequency of DSP diagnosed by the monofilament than by the sum of NSS and NDS, combined with the simplicity of the former technique, impels us to suggest the Semmes–Weinstein monofilament as an appropriate tool to diagnose DSP in the primary care setting in Brazil and, possibly, in other countries with similar social and economic characteristics, but further studies are needed to confirm this finding. The sum of NSS and NDS is time-consuming and too complex to be adopted in general practice. After comparing the vibration threshold, 10 g monofilament, temperature perception and the modified Michigan Neuropathy Screening Instrument questionnaire in English for individuals with T2D followed in primary care, Rahman et al. considered the Semmes–Weinstein monofilament as the more appropriate, cheaper and easier tool for assessing DSP in general practice [18].

The prevalence of definitive CAN was 10%, similar to observations in recent studies which evaluated individuals with T2D followed in primary care in Saudi Arabia (15.3%) [19] and Denmark (9% and 15% in two different time points, seven years apart) [20], although the diagnosis was performed using different criteria. The frequency of metformin use and of hypercholesterolemia, and total cholesterol concentrations, were significantly different between the groups with and without definitive CAN. Other variables, however, presented a borderline *p* value: longer diabetes duration, a known risk factor for the development of CAN [10], arterial hypertension and higher triglycerides concentrations. These (and other) cardiovascular risk factors have been associated with CAN in individuals with T2D [19,21,22,23,24], and highlight the importance of addressing not only glycemic control but also hypertension and dyslipidemia as a preventive strategy against CAN [25].

The main strengths of this study are the relatively large size of the sample evaluated by a highly specific method for CAN diagnosis (CARTs combined with spectral analysis of the HRV) [10] and the possible applicability of the results, which suggest the Semmes–Weinstein monofilament as a tool for DSP screening in individuals with T2D followed in primary health care units in Brazil. The main limitations are (1) those associated with the cross-sectional design, such as the “incidence-prevalence” bias, considering that T2D is a longer-lasting disease and that risk factors resulting in death may be under-represented; (2) the absence of data regarding retinopathy and albuminuria, the latter not systematically analyzed in this population until the beginning of this study; and (3) the limited generalizability of the findings, given that the government-subsidized T2D treatment program provides sulphonylureas, metformin and insulin, but not other classes of medicines that could positively impact diabetes complications, such as glucagon-like peptide 1 (GLP1) receptor agonists, dipeptidyl peptidase 4 (DPP4) inhibitors and sodium–glucose cotransporter 2 (SGLT2) inhibitors.

Although the present findings cannot necessarily be extrapolated to other primary care units, they show that those 14.3% presenting DSP and 10% presenting CAN are at high risk of minor amputations [26] and of major cardiovascular events [5], respectively, despite being followed in a low complexity center.

## 5. Conclusions

In the primary care setting, the prevalence of DSP in Brazilian adults with T2D was low (6.3%) as evaluated by the sum of NSS and NDS, and was 14.3% as evaluated by the Semmes–Weinstein monofilament. The prevalence of definitive CAN was 10% as evaluated by CARTs combined with spectral analysis of the HRV.

## Figures and Tables

**Figure 1 ijerph-17-03232-f001:**
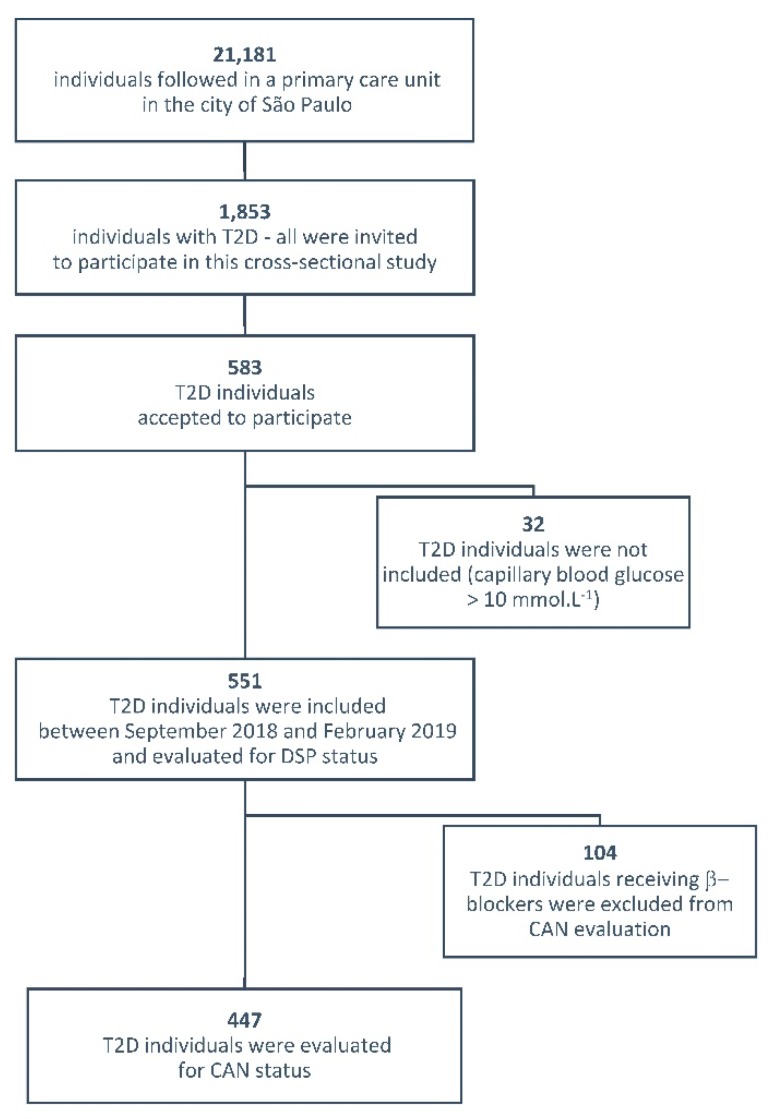
Flow diagram showing the recruitment of type 2 diabetes (T2D) individuals in the study. DSP: Distal symmetric polyneuropathy; CAN: Cardiovascular autonomic neuropathy.

**Table 1 ijerph-17-03232-t001:** Characteristics of individuals with type 2 diabetes according to the status of distal symmetric polyneuropathy (DSP) as evaluated by the monofilament.

Demographic, Clinical and Biochemical Characteristics	All Individuals (*n* = 551)	Without DSP by Monofilament (*n* = 472)	With DSP by Monofilament (*n* = 79)	*p*-Value
Age (years)	65 (59–72)	65 (59–72)	65 (58–73)	0.97
Sex (% female)	59.3	60	53	0.24
Ethnicity (C/N/A) (%)	68/31/1	67.7/31.6/0.7	67.0/30.4/2.6	0.18
Height (cm)	161 (154–169)	**161 (154–168)**	**165 (156–173)**	**0.008**
Body mass index (kg/m^2^)	29 (25.8–33.2)	29.0 (25.8–32.9)	29.9 (25.1–34.7)	0.35
Waist circumference (cm)	102 (94–112)	102 (94–111)	103 (92–118)	0.42
Arterial hypertension (%)	72	71	77	0.30
Smoking (%)	10	10	10	0.96
eGFR (mL.min^−1^.1.73 m^2^)	77 (59.6–92.6)	77.0 (60.0–92.0)	77.0 (59.0–93.0)	0.74
Total cholesterol (mmol·L^−1^)	5.04 (4.24–5.76)	5.06 (4.26–5.76)	4.68 (4.08–5.5)	0.14
HDL (mmol·L^−1^)	1.2 (0.96–1.4)	1.2 (0.98–1.4)	1.1 (0.85–1.5)	0.31
LDL (mmol·L^−1^)	2.9 (2.3–3.6)	3.0 (2.3–3.6)	2.9 (2.4–3.5)	0.59
Triglycerides (mmol·L^−1^)	1.8 (1.3–2.5)	1.8 (1.3–2.5)	1.8 (1.2–2.6)	0.47
Hypercholesterolemia (%)	72	73	67	0.30
**Diabetes status**				
Diabetes duration (years)	10 (5–15)	**8.0 (4.0–15.0)**	**13.0 (6.0–21.0)**	**0.0008**
HbA_1_C (mmol·L^−1^)	55 (45–76)	**54 (44–75)**	**65 (48–85)**	**0.02**
eGFR < 60 mL.min^−1^.1.73 m^2^ (%)	23.5	23	23	0.84
DSP by NSS and NDS (%)	6.3	**2**	**33**	**<0.0001**
DSP by monofilament (%)	14.3	-	-	-
Amputation (%)	1.6	**0**	**11.4**	**<0.0001**
Incipient CAN (%)	12.5	10.0	11.4	0.91
Definitive CAN (%)	10	12.7	11.4	0.91
Metformin (%)	75	76	68	0.13
Sulphonylureas (%)	35.3	36	38	0.32
NPH insulin (%)	25	**23**	**41**	**0.0006**
Regular Insulin (%)	8.3	**6**	**19**	**0.0001**
Statins (%)	31	32	32	0.91
ACEI (%)	29	29	33	0.48
ARB (%)	29.2	29	25	0.49
Beta-blockers (%)	18.5	19	18	0.74
Fibrates (%)	4.5	4	5	0.80

Data expressed as median (interquartile range). Bold: *p* < 0.05 between the groups with and without DSP. ACEI: Angiotensin-converting-enzyme inhibitors; ARB: angiotensin II receptor blockers; CAN: cardiovascular autonomic neuropathy; DSP: distal symmetric polyneuropathy; eGFR: estimated glomerular filtration rate; HDL: High-density lipoprotein; LDL: Low-density lipoprotein; NDS: Neuropathy Disability Score; NPH: Neutral Protamine Hagedorn; NSS: Neuropathy Symptoms Score; Ethnicity (C: Caucasoid; N: Negroid; A: Asiatic); Hypercholesterolemia: LDL > 2.6 mmol/L (100 mg/dL) or use of statin. The missing data (percentage) for each reported variable are as follows: Age (0%); Sex (0%); Ethnicity (0%); Height (0%); Body mass index (6.4%); Waist circumference (13.6%); Arterial hypertension (0%); Smoking (0%); eGFR (11.2%); Total cholesterol (11.2%); HDL (11.2%); LDL (11.2%); Triglycerides (11.2%); Hypercholesterolemia (0%); Diabetes duration (2.7%); HbA1C (11.2%); LPA (21.6%); DSP by NSS and NDS (0%); DSP by monofilament (0%); Amputation (0%).

**Table 2 ijerph-17-03232-t002:** Characteristics of individuals with type 2 diabetes according to the status of cardiovascular autonomic neuropathy (CAN).

Demographic, Clinical and Biochemical Characteristics	All Individuals (*n* = 447)	Individuals without Definitive CAN (*n* = 408)	Individuals with Definitive CAN (*n* = 39)	*p* Value
Age (years)	65 (59–72)	65 (59–72)	62 (56–70)	0.33
Sex (% female)	59.1	59	56	0.72
Ethnicity (C/N/A) (%)	68/31/1	68.0/31.0/1.0	62.0/33.0/5.0	0.36
Height (cm)	162 (154–169)	162 (155–169)	162 (152–166)	0.56
Body mass index (kg/m^2^)	29 (25.7–32.9)	28.9 (25.6–32.9)	31.5 (26.4–34.2)	0.10
Waist circumference (cm)	102 (94–112)	102 (94–112)	103 (94–112)	0.86
Arterial hypertension (%)	66	65	80	0.06
Smoking (%)	10	10.5	7.7	0.57
eGFR (mL.min^−1^.1.73 m^2^)	78 (60.5–93.6)	79 (60.5–93.6)	75 (60.5–90.8)	0.32
Total cholesterol (mmol·L^−1^)	5.0 (4.3–5.8)	**5.0 (4.3–5.8)**	**5.3 (4.9–6.0)**	**0.03**
HDL mmol·L-^1^)	1.2 (1.1–1.4)	1.2 (1.1–1.4)	1.2 (0.9–1.4)	0.70
LDL (mmol·L-^1^)	3.0 (2.4–3.7)	3.0 (2.4–3.7)	3.3 (2.6–4.0)	0.13
Triglycerides (mmol·L^−1^)	1.8 (1.3–2.5)	1.8 (1.2–2.4)	2.1 (1.5–2.7)	0.07
Hypercholesterolemia (%)	70	**68**	**84**	**0.03**
**Diabetes status**				
Diabetes duration (years)	8 (4–15)	8 (4–15)	11 (6–18)	0.08
HbA_1_C (mmol·L^−1^)	55 (45–76)	54 (45–76)	58 (50–84)	0.14
eGFR < 60 mL.min^−1^.1.73 m^2^ (%)	21.8	22	21	0.92
DSP by NSS and NDS (%)	6.9	6.6	10.3	0.39
DSP by monofilament (%)	14.6	14.3	18.0	0.53
Amputation (%)	2	2	2.3	0.56
Metformin (%)	75.8	**74**	**92**	**0.01**
Sulphonylureas (%)	36	35	46	0.16
NPH insulin (%)	23	23	26	0.66
Regular Insulin (%)	7.6	8	2.6	0.21
Statins (%)	28	26	41	0.05
ACEI (%)	28.6	27	38	0.15
ARB (%)	28.2	27	36	0.26
Fibrates (%)	4.3	4	10	0.05

Data expressed as median (interquartile range). Bold: *p* < 0.05 between the groups with and without CAN. ACEI: Angiotensin-converting-enzyme inhibitors; ARB: angiotensin II receptor blockers; CAN: cardiovascular autonomic neuropathy; DSP: distal symmetric polyneuropathy; eGFR: estimated glomerular filtration rate; HDL: High-density lipoprotein; LDL: Low-density lipoprotein; NDS: Neuropathy Disability Score; NPH: Neutral Protamine Hagedorn; NSS: Neuropathy Symptoms Score; Ethnicity (C: Caucasoid; N: Negroid; A: Asiatic); Hypercholesterolemia: LDL > 2.6 mmol/L (100 mg/dL) or use of statin. The missing data (percentage) for each reported variable are as follows: Age (0%); Sex (0%); Ethnicity (0%); Height (0%); Body mass index (6.5%); Waist circumference (14.9%); Arterial hypertension (0%); Smoking (0%); eGFR (12%); Total cholesterol (12%); HDL (12%); LDL (12%); Triglycerides (12%); Hypercholesterolemia (0%); Diabetes duration (2.5%); HbA1C (12%); LPA (22.8%); DSP by NSS and NDS (0%); DSP by monofilament (0%); Amputation (0%).

**Table 3 ijerph-17-03232-t003:** Multivariate analysis of risk factors for distal symmetric polyneuropathy (DSP) and for cardiovascular autonomic neuropathy (CAN).

Risk Factors	Odds Ratio	95% Confidence Interval	*p* Value
**DSP by Monofilament**			
Male sex	1.05	0.58–1.90	0.864
Age	0.99	0.96–1.02	0.734
**Diabetes duration**	**1.05**	**1.02–1.08**	**0.0003**
HbA1c	1.07	0.94–1.21	0.275
Arterial hypertension	1.09	0.52–2.27	0.813
Smoking	0.77	0.28–2.10	0.601
Cholesterol	0.99	0.98–1.00	0.146
Triglyceride	1.00	0.99–1.00	0.662
HDL-Cholesterol	0.99	0.97–1.01	0.643
Waist circumference	1.01	0.99–1.03	0.050
**CAN**			
Male sex	0.89	0.39–2.04	0.796
Age	0.98	0.94–1.02	0.417
Diabetes duration	1.01	0.96 - 1.05	0.631
HbA1c	1.01	0.93–1.29	0.265
**Arterial hypertension**	**3.25**	**1.12–9.39**	**0.016**
Smoking	1.11	0.30–4.04	0.872
**Cholesterol**	**1.01**	**1.0007–1.02**	**0.031**
Triglyceride	1.00	0.99–1.00	0.683
HDL-Cholesterol	0.99	0.96–1.02	0.687
Waist circumference	0.98	0.95–1.00	0.191

HDL: High-density lipoprotein; NDS: Neuropathy Disability Score; NSS: Neuropathy Symptoms Score. Bold: *p* < 0.05.

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
