# Peer review of "Distal Symmetric and Cardiovascular Autonomic Neuropathies in Brazilian Individuals with Type 2 Diabetes Followed in a Primary Health Care Unit: A Cross-Sectional Study"

_ijerph, 2020, doi:10.3390/ijerph17093232_

Round 1
Reviewer 1 Report
Comments:
1. Methods, section: What criteria the authors used to define normal/abnormal values of the CARTs and spectral analysis? The authors should make a statement together with reference (s) to this
2. Regression analyses would be more useful and informative to look for independent associations between DSP or CAN with the variables examined.
I suggested major revision.
Author Response
Reviewer 1
Comments:
1. Methods, section: What criteria the authors used to define normal/abnormal values of the CARTs and spectral analysis? The authors should make a statement together with reference (s) to this.
This statement is located in the first paragraph of page 3 (reference # 12).
2. Regression analyses would be more useful and informative to look for independent associations between DSP or CAN with the variables examined.
Thank you for the suggestion. We inserted a Table (Table 3) showing multivariate analyses of the risk factors for DSP diagnosed by monofilament and for CAN. We decided not to show detailed data for DSP diagnosed by the sum of NSS and NDS due to the low number of patients (n=35) that received this diagnosis by this methodology, which could result in false-positive results.
Reviewer 2 Report
The authors aim to evaluate the prevalence rates of DSP and CAN in Brazilian adults with T2D in a primary health care unit. A large number of individuals (551) were recruited for the study. Although there was a low prevalence of DSP as evaluated by the sum of NSS and NDS (6.3%), and by the Semmes-Weinstein monofilament (14.3%), the results suggested that the use of monofilament may be an applicable tool to diagnose DSP in the primary care setting in Brazil. In addition, the authors suggested that the prevalence of definitive CAN was 10% as evaluated by CARTs combined with spectral analysis of the HRV. However, the article still is open to criticisms.
Major comments:
- The major findings in the article are already common knowledge; there are no new findings.
- The authors presented results based on their study, and discussed their results compared to other studies. However, no unique findings were proposed in their conclusion.
- Since cross-sectional survey was performed in this study, authors should point out the time of data collection.
- Preparation of the Tables is not good; number, unit, and arrangement should be well-organized.
- The method of estimated glucose disposal rate (eGDR) is a marker for the assessment of insulin resistance of type 1 DM, which is not ideal for the study.
- There were some inconsistency in data presented between texts and tables, eg. higher frequency of DSP diagnosed by monofilament (10.3 vs 74%)
- The authors concluded that the higher “prevalence rate” of DSP with the use of monofilament suggests it may be a more appropriate tool than NSS+NDS to diagnose DSP. For chronic disease studies or studies of long-lasting risk factors, prevalence odds ratio (POR) is the preferred measure of association than prevalence ratio (PR) in cross-sectional studies, especially in higher prevalence diseases (more than 10-30%). Besides, if conclusion and suggestion were proposed, authors should present the comparison results in a statistical significance.
Minor comments:
- It would be appreciated if authors include P value in Table 1, if possible.
- Total patients number that diagnosed by monofilament as with or without DSP was not 551 in table 1.
- In Table 1, “DSP by NDS and NSS (%)” could be amended to “DSP by NSS and NDS (%)” to make it coherent in the text.
- In Table 1, “(%)” should be added after Incipient CAN.
- Reference format should follow the rule of the Journal, International Journal of Environmental Research and Public Health.
Author Response
Reviewer 2
Major comments:
1. The major findings in the article are already common knowledge; there are no new findings.
You are right. The only new findings of the present study are the prevalence rates of CAN and of DSP in the context of primary care in the biggest Brazilian city. We consider that this manuscript could fit in the Special issue “The Burden of Diabetes on Health Services”, given (1) the high prevalence of T2D in Brazil (16.8 millions, according to IDF); (2) the paucity of epidemiological data regarding diabetes complications in Brazil, especially in the primary care setting and (3) the implications of DSP and of CAN for amputations and major cardiovascular events, respectively. We do believe that the findings of this study have the potential of assisting decision-makers in understanding the magnitude of the problem in Brazil.
2. The authors presented results based on their study, and discussed their results compared to other studies. However, no unique findings were proposed in their conclusion.
Please, refer to the previous (#1) comment.
3. Since cross-sectional survey was performed in this study, authors should point out the time of data collection.
Time of data collection was inserted in section 2.1.
4. Preparation of the Tables is not good; number, unit, and arrangement should be well-organized.
We maintained only International System of Units and replaced some names by abbreviations in an attempt to improve the comprehension of the tables. We are not showing detailed data for DSP diagnosed by the sum of NSS + NDS due to the low number of patients (n=35) that received this diagnosis by this methodology, which could result in false-positive results.
5. The method of estimated glucose disposal rate (eGDR) is a marker for the assessment of insulin resistance of type 1 DM, which is not ideal for the study.
Despite the existence of manuscripts using this surrogate marker of insulin resistance in type 2 diabetes, we followed the opinion of this referee and deleted data regarding eGDR.
6. There were some inconsistency in data presented between texts and tables, eg. higher frequency of DSP diagnosed by monofilament (10.3 vs 74%)
We deleted this paragraph because we decided not to show detailed data for DSP diagnosed by the sum of NSS and NDS due to the low number of patients (n=35) that received this diagnosis by this methodology. However, these numbers were correct – we inserted the word “respectively” inside all the parenthesis, because the phrase is: In comparison to individuals without DSP as evaluated by monofilament, those with DSP were taller (161 [without DSP] versus 165 cm [with DSP], respectively); presented longer diabetes duration (8 [without DSP] versus 13 [with DSP] years, respectively) etc.
7. The authors concluded that the higher “prevalence rate” of DSP with the use of monofilament suggests it may be a more appropriate tool than NSS+NDS to diagnose DSP. For chronic disease studies or studies of long-lasting risk factors, prevalence odds ratio (POR) is the preferred measure of association than prevalence ratio (PR) in cross-sectional studies, especially in higher prevalence diseases (more than 10-30%). Besides, if conclusion and suggestion were proposed, authors should present the comparison results in a statistical significance.
Thank you for the suggestion. We inserted a Table (Table 3) showing the prevalence odds ratios of the risk factors for DSP diagnosed by monofilament and for CAN. We decided not to show detailed data for DSP diagnosed by the sum of NSS and NDS due to the low number of patients (n=35) that received this diagnosis by this methodology, which could result in false-positive results.
We suggested, rather than concluded, that monofilament may be a better tool than the sum of NSS and NDS in Brazil and in countries with similar realities. As we do not have a third "gold standard" method for diagnosing DSP with which we could compare the performance of monofilament and of NSS + NDS, we were unable to perform a statistical analysis. However, we inserted, in the Discussion section, a phrase stating that additional studies are needed to confirm these findings.
Minor comments:
1. It would be appreciated if authors include P value in Table 1, if possible. Total patients number that diagnosed by monofilament as with or without DSP was not 551 in table 1.
Thank you. The P values were inserted in the Table and the number of patients was corrected.
2. In Table 1, “DSP by NDS and NSS (%)” could be amended to “DSP by NSS and NDS (%)” to make it coherent in the text.
Thank you for the observation. We decided to remove data regarding DSP diagnosed by the sum of NSS and NDS from Table 1 due to the low number of patients (n=35) that received this diagnosis by this methodology, which could result in false-positive results.
3. In Table 1, “(%)” should be added after Incipient CAN.
Thank you – the symbol was inserted.
4. Reference format should follow the rule of the Journal, International Journal of Environmental Research and Public Health.
Thank you. The format was modified accordingly.
Reviewer 3 Report
The authors conducted a crossed-sectional study about the prevalence of distal symmetric and cardiovascular autonomic neuropathies in adults with T2DM. My concerns:
- The authors have not conformed to the STROBE STATEMENT for cross-sectional studies.
- The results of the study do not seem to add much to the current medical literature and they are rather contradictory.
- The sample is relatively small for a cross-sectional study.
Author Response
Reviewer 3
1. The authors have not conformed to the STROBE STATEMENT for cross-sectional studies.
Thank you for the suggestion. We included items required by the STROBE Statement that were missing: how the study size was arrived at with a flow diagram (Figure 1), number of participants with missing data (Table 1 footnote) etc. Please, find attached The Strobe Statement.
2. The results of the study do not seem to add much to the current medical literature and they are rather contradictory.
The only new findings of the present study are the prevalence rates of CAN and of DSP in the context of primary care in the biggest Brazilian city. We consider that this manuscript could fit in the Special issue “The Burden of Diabetes on Health Services”, given (1) the high prevalence of T2D in Brazil (16.8 millions, according to IDF); (2) the paucity of epidemiological data regarding diabetes complications in Brazil, especially in the primary care setting and (3) the implications of DSP and of CAN for amputations and major cardiovascular events, respectively. We do believe that the findings of this study have the potential of assisting decision-makers in understanding the magnitude of the problem in Brazil.
3. The sample is relatively small for a cross-sectional study.
There are not so many cross-sectional studies with large samples that have performed the systematic analysis of CAN by CARTs and spectral analysis, especially in the context of primary care. We only found two studies conducted in primary care centers, with which we compared our results: AlOlaiwi et al. in Saudi Arabia (n = 400) and Andersen et al. in Denmark (n = 777 in the first evaluation and n = 443 in the second evaluation, 7 years apart). Thus, the present series of 447 individuals is not that small considering the relative complexity of CAN diagnosis and the fact that the study was conducted in a primary care unit.

Round 2
Reviewer 1 Report
The authors performed the requested changes and corrections.
Author Response
The authors performed the requested changes and corrections.
Thank you very much for the suggestions.
Reviewer 3 Report
The authors have tried to address to the Reviewer's concerns. A few more details should be taken into consideration:
- The authors are kindly requested to mention in the Title the type the study they conducted.
- The authors are kindly requested to explain in a more thorough way the scientific background and rationale for the investigation being reported and to in the Introduction section and to state specific objectives, including any prespecified hypotheses.
- All numbers should be eliminated from the Methods section and the authors are requested to provide: bias and study size calculations
- The authors are kindly requested to provide elements of their statistical analysis tests in the results section.
- The authors are kindly requested to discuss the limitations of their study in the Discussion section.
Author Response
1. The authors are kindly requested to mention in the Title the type the study they conducted.
We inserted “a cross-sectional study” in the title.
2. The authors are kindly requested to explain in a more thorough way the scientific background and rationale for the investigation being reported and to in the Introduction section and to state specific objectives, including any prespecified hypothese
We inserted a paragraph in the Introduction section better explaining the rationale for the present study. The objective of the study was descriptive, to estimate the prevalence of both neuropathic complications in this population, for the purposes of public health planning. Thus, there was no prespecified hypothesis.
3. All numbers should be eliminated from the Methods section and the authors are requested to provide: bias and study size calculations
Sorry, we thought the question of the sample size was clear in the flow chart elicited in Figure 1 and in the first phrase of the Methods section: among the 21,181 individuals followed in the primary care unit, 1,853 had type 2 diabetes and all were invited to participate in the study, but only 583 accepted to participate. Thus, this study was conducted in a convenience sample, since there was no sample size calculation – we tried to include as much individuals with T2D as possible. We rephased the text in an attempt to make it clearer.
We inserted a phrase about the bias related to the cross-sectional design of the study in the end of Discussion section (limitations).
4. The authors are kindly requested to provide elements of their statistical analysis tests in the results section.
We inserted the P values along the text.
5. The authors are kindly requested to discuss the limitations of their study in the Discussion section.
The limitations are being presented in the end of Discussion, from lane 218 to 227.